# Evidence and Therapeutic Perspectives in the Relationship between the Oral Microbiome and Alzheimer’s Disease: A Systematic Review

**DOI:** 10.3390/ijerph182111157

**Published:** 2021-10-24

**Authors:** Yoann Maitre, Rachid Mahalli, Pierre Micheneau, Alexis Delpierre, Gilles Amador, Frédéric Denis

**Affiliations:** 1Emergency Department, Montpellier University Hospital, 34090 Montpellier, France; maitreyoann@yahoo.fr; 2EA 2415, Aide à la Décision pour une Médecine Personnalisée, Université de Montpellier, 34093 Montpellier, France; 3Department of Odontology, Tours University Hospital, 37000 Tours, France; rachid.mahalli@etu.univ-tours.fr (R.M.); pierre45892@gmail.com (P.M.); alexis.delpierre@outlook.fr (A.D.); 4Faculty of Dentistry, Nantes University, 44000 Nantes, France; gilles.amador@univ-nantes.fr; 5EA 75-05 Education, Ethique, Santé, Faculté de Médecine, Université François-Rabelais, 37044 Tours, France

**Keywords:** Alzheimer’s disease, oral microbiota, oral microbiome, mental disorders

## Abstract

This review aims to clarify the nature of the link between Alzheimer’s disease and the oral microbiome on an epidemiological and pathophysiological level, as well as to highlight new therapeutic perspectives that contribute to the management of this disease. We performed a systematic review, following the Preferred Reporting Items for Systematic Reviews checklist, from January 2000 to July 2021. The terms “plaque,” “saliva,” and “mouth” were associated with the search term “oral diseases” and used in combination with the Boolean operator “AND”/“OR”. We included experimental or clinical studies and excluded conferences, abstracts, reviews, and editorials. A total of 27 articles were selected. Evidence for the impact of the oral microbiome on the pathophysiological and immunoinflammatory mechanisms of Alzheimer’s disease is accumulating. The impact of the oral microbiome on the development of AD opens the door to complementary therapies such as phototherapy and/or the use of prebiotic compounds and probiotic strains for global or targeted modulation of the oral microbiome in order to have a favourable influence on the evolution of this pathology in the future.

## 1. Introduction

Alzheimer’s disease (AD) is progressive and neurodegenerative in the elderly (>65 years old), in which the main symptom is a decline in cognitive ability, severe enough to interfere with daily living activities. Approximately 47 million people worldwide have AD-related dementia, and this number has doubled almost every 20 years. It is expected to reach 131.5 million by 2050 [1].

AD is characterized by protein conformation [2,3], mainly caused by aberrant processing and polymerization of normally soluble proteins [4]. When misfolded, soluble neuronal proteins reach altered conformations, due to genetic mutations (Apolipoprotein ε4 gene (APO ε4) is the most important risk factor in sporadic AD), external factors like aging and aggregates, lead to abnormal neuronal function and neuronal loss [2,3,4,5]. To date, AD causes are still poorly understood and, although many treatments have been tested in clinical trials, none can cure or stop its progression.

However, it was established that even before the first symptoms appear, neurons are affected by two types of lesions: amyloid plaques located between neurons and neurofibrillary degeneration, which is located inside neurons [6]. Amyloid plaques are formed by the abnormal accumulation of a protein called “β-amyloid” between nerve cells in the grey matter of the cerebral cortex, with a dysfunction of the connections between neurons [7]. Neurofibrillary degeneration corresponds to an abnormal accumulation of filaments inside the neuron; the protein responsible for this dysfunction is called “Tau protein” [8]. Both lesions are clusters of proteins that form during the normal aging process. Yet, in Alzheimer’s type disease, these proteins accumulate in much larger quantities [9].

AD is considered a multifactorial disease, with its onset thought to result from the interaction between genetic background and various risk factors [10]. Age is the main proven risk factor, with a prevalence that doubles every 5 years from the age of 65 years (2% after 65, 15% after 80 years) [1]. Other risk factors are now well-established, such as low level of education, cardiovascular risk factors (untreated hypertension, stroke, hypercholesterolemia, diabetes, overweight status, obesity, and periodontal disease); women; environmental factors (tobacco, alcohol, pollution, certain drugs); and sleep disorders. Chronic inflammation of the body is linked to shrinkage of brain areas in AD; mood disorders like chronic stress or depression are also correlated. An unbalanced diet, lack of physical and stimulating mental activity may also be associated with an increased risk of AD [11,12,13].

Recently, many studies show epidemiological correlations between AD and periodontitis. There are also pathophysiological arguments for a causal relationship between AD and periodontal disease (PD) [14,15]. Periodontitis is a peripheral infectious/inflammatory condition, one of the major risk factors in tooth loss [16]. Periodontitis is associated with increased serum levels of C-reactive protein (CRP) and pro-inflammatory cytokines (e.g., tumor necrosis factor-α), as well as decreased anti-inflammatory markers (e.g., interleukin-10) [17]. Many researchers hypothesised that the association between AD and periodontitis may be due to increased systemic inflammation accompanying the growth of periodontal pathogens [18,19]. The latter would mediate the development of AD through their role in the development of vascular disease [14,20,21].

Although the link between gut dysbiosis and neurological disorders remains speculative, anatomical evidence is leading to a better understanding of a bidirectional link between the gut and brain, as is the case in AD [22,23]. This bidirectional relationship is thought to exist between the oral microbiota and many systemic diseases, including AD.

This review aims to clarify the nature of the link between Alzheimer’s disease and the oral microbiome at the epidemiological and pathophysiological levels, as well as to highlight new therapeutic perspectives to contribute to the management of this disease.

## 2. Materials and Methods

We performed a systematic review following the Preferred Reporting Items for Systematic Reviews (PRISMA) checklist [24]. Authors defined research questions, objectives, search strategies, and inclusion/exclusion criteria through previous discussions.

### 2.1. Search Strategy

Understanding the interactions of microorganisms with the host and with each other, as well as their impact on health, was improved by the development of high-throughput sequencing techniques for genetic material since 2000 [25]. The research was carried out from January 2000 to July 2021 to clarify the relationship between AD and the oral microbiome. So as not to restrict the search only to oral pathologies, terms identifying the main oral ecological niches “Dental plaque,” “Saliva,” and Mouth” were associated with the search term “Mouth diseases” and used in combination with the Boolean operator “AND”/”OR” as per the following equation: 

((((“Mouth”[Mesh]) OR “Mouth Diseases”[Mesh]) OR “Dental Plaque”[Mesh]) OR “Saliva”[Mesh]) AND “Alzheimer Disease”[Mesh]

The search was performed in the MEDLINE literature database through PubMed interface.

### 2.2. Inclusion and Exclusion Criteria

We included experimental or clinical studies (longitudinal, cross-sectional, or randomised) that explored the association between the oral microbiome and AD symptoms, and/or explored the association between the oral microbiome and AD from an immuno-inflammatory perspective. We excluded conferences, abstracts, reviews, and editorials.

### 2.3. Data Extraction

Two independent operators (F.D. and Y.M.) manually checked the references of eligible studies on the subject, according to inclusion/exclusion criteria.

An analysis of titles and abstracts by F.D. and Y.M. was performed to identify potentially relevant articles based on inclusion criteria. When the information in the abstracts was deemed insufficient, a review of the complete study decided on its relevance. A discussion between the two reviewers took place to reach a consensus with discrepancies in the selection decision process.

## 3. Results

### 3.1. Study Selection

After retrieving the studies from the database, we identified the studies that met eligibility criteria. Of the 227 studies initially identified, 24 were selected. A complementary detection resulted in the selection of three additional articles. A total of 27 articles were included in the analysis (Figure 1).

### 3.2. Study Characteristics

Since the beginning of 2000, the number of publications about the relationship between the oral microbiome and AD has been steadily increasing, with 44% of publications from 2019 to 2021 (Figure 2a). These concern experimental in vitro studies (n = 1) or murine models (n = 6) and human clinical studies (n = 20). The case-control study design is most often used for human studies (n = 12) (Figure 2b). With the exception of retrospective cohort studies, sample sizes were limited in both animal (N = 30 to 140) and human (N = 20 to 354) studies. While they were conducted worldwide, they were concentrated in the United States (9 studies from 2002 to 2021) and in Europe (8 studies from 2010 to 2021) see Figure 2c.

Each study meeting the inclusion criteria was analysed in terms of authors, date of publication, study design, objectives, factors assessed, results and limitations (Table 1).

### 3.3. Clinical Oral Perturbations and Alzheimer’s Disease

In 2020, Kantarci et al. showed that PD increases bone loss in AD model mice (5xFAD) and wild-type control mice (WT). While they found no significant difference in alveolar bone loss after induction of experimental PD between 5xFAD and WT mice, they noted that alveolar bone loss is higher in 5xFAD than in WT at baseline [26].

Periodontal damages were also identified in several case-control studies in humans. Thus, De Oliveira Araújo et al. found a significant association between periodontitis and AD, controlling for age, sex, income, and education [27]. In accordance with this result, periodontal infections, the presence of deep periodontal pockets, and generalised marginal alveolar bone loss are markers of periodontal deterioration, that appear to be significantly associated with AD patients [28,29,30,31]. Similarly, in 2015, the cross-sectional study by Kamer et al. found a significant association between a loss of clinical attachment and increased amyloid load in vulnerable brain areas of AD patients, after adjustment for confounding factors (age, ApoE, and smoking) [15].

Several retrospective cohort studies, assessing the association between incident AD risk and PD, moderate these results. For example, in a cohort followed for 26 years (N = 6650), the association between risk of AD and periodontal pocket depth appeared marginal in men and older individuals [32]. The study by Choi et al. shows that in a cohort of 262,349 participants aged 50 or older with a follow-up of 11 years, developing AD tends to be higher in patients with chronic periodontitis (CP), vs. patients without CP (aHR = 1.05; 95% CI = 1.00–1.11, *p* = 0.042) [21]. Conversely, in a cohort of 27,963 patients over 50 years, followed for 16 years, Chen et al. showed a significantly higher risk of developing AD in patients with 10 years of exposure to CP vs. those with no exposure [33].

### 3.4. Oral Dysbiosis and Alzheimer’s Disease

Liu et al. showed that the salivary microbiome has significantly lower diversity in patients with AD than in healthy patients [34]. Maurer et al. also identified a higher bacterial load of *Aggregatibacter actinomycetemcomitans*, *Porphyromonas gingivalis*, and *Fusobacterium nucleatum* in dental plaque compared to controls [35]. Rivière et al. demonstrated that brain colonisation by treponemes (*T. amylovorum*, *T. denticola*, *T. maltophilum*, *T. medium*, *T. pectinovorum*, *T. socranskii*, *T. vincentii*) was more frequent in AD patients than controls [36]. In 2018, Ilievski et al. detected the presence of *P. gingivalis* and gingipaïn in the hippocampus of mice [37].

While the study by Liu et al. concluded that no oral bacteria are associated with the severity of AD [34], serum immunoglobulin G (IgG) concentration associated with common periodontal bacteria (*P. gingivalis*, *Tannerella forsythia*, *A. actinomycetemcomitans*, *T. denticola*, *Campylobacter rectus*, *Actinomyces naeslundii)* represents a risk factor in developing AD [38]. According to the study by Beydoun et al., the incidence and mortality of AD could be related to a combination of periodontal bacteria. The incidence of AD is related to a composite of *C. rectus* and *P. gingivalis*, such that the risk of mortality from AD was enhanced by a composite of *P. gingivalis*, *Prevotella intermedia*, *Prevotella nigrescens*, *F. nucleatum*, *C. rectus*, *Streptococcus intermedius*, *Capnocytophaga ochracea and Prevotella melaninogenica* [32]. 

### 3.5. Pathophysiological Evidence between the Microbiome and Alzheimer’s Disease

According to the study of Dominy et al., on mice, the presence of *P. gingivalis* and gingipaïn in the brain plays a central role in the pathogenesis of AD [39]. In vitro, Yamada et al. demonstrated that exposure to phosphoglycerol dihydroceramide produced by *P. gingivalis* (PGDHC) increased Aβ peptide secretion in a dose-dependent manner in the Chinese hamster ovary-7WD10 cells, stably expressing wild-type human amyloid precursor protein (APP) and induced specific phosphorylation of Tau protein in a dose-dependent manner in human neuroblastoma cells [40]. In rats, Zhang et al. showed a significant elevation of Aβ1-40 peptide levels in the cerebral cortex of AD rats with periodontitis compared to AD rats without it [41]. Similarly, Díaz-Zúñiga et al. observed a significant increase in Aβ1-42 levels and Tau hyperphosphorylation in the hippocampus of rats orally infected by K1 or K2 *P. gingivalis*, associated with PD [42]. In mice, Kantarci et al. noted a significant elevation of the mean level of insoluble Aβ42 peptide in the prefrontal cortex of AD mice with periodontitis, compared to AD mice without it [26]. Phosphorylation of Tau protein was detected by Ilievski et al. (2018) in AD mice infected with *P. gingivalis*, but not in control mice [37].

Studies of cognitive decline in murine models show no significant differences in learning and memory tests (Y maze, nest building, or the Morris Water Maze test) that may be associated with periodontitis or acute or continuous exposure to *P. gingivalis*-derived lipopolysaccharides (Pg-LPS) [41,43]. According to Ide et al., there was also no significant relationship between serum levels of antibodies to *P. gingivalis* and rates of cognitive decline, based on the Mini Mental State Examination (MMSE) in AD patients [19]. However, in 2020, Leblhuber et al. found a significant association between the salivary presence of *P. gingivalis* and lower MMSE in AD patients [44].

### 3.6. Oral Microbiota and Immuno-Inflammation in Alzheimer Disease

While neuroimmune interactions are essential to brain function [45], chronic inflammation appears to be a key factor in the pathophysiology and clinical progression of AD [46]. In vitro, Yamada et al. showed that exposure of human neuroblastoma cells to PGDHC or Pg-LPS increased expression of the senescence-associated secretory phenotype involving β-galactosidase, cathepsin B, cysteine, and pro-inflammatory cytokines TNF-α and IL-6 [40]. 

In the hippocampus of rats infected with *P. gingivalis* encapsulated serotypes K1 and K2, Díaz-Zúñiga et al. observed a significant increase in pro-inflammatory cytokine (IL-1β, IL-6, TNF-α, IFN-γ) not found in rats infected with non-encapsulated bacterial strains [42]. Similarly, oral application of *P. gingivalis* 3 times a week for 22 weeks showed a significant increase in the expression of pro-inflammatory cytokines (IL-1β, IL-6 and TNF-α) in the hippocampus of infected mice compared to controls [37]. In AD model rats with induced periodontitis, compared to controls, TNF-α levels were significantly higher in the hippocampus, IL-1 levels were higher in the cerebral cortex, and IL6 levels were higher both in the hippocampus and cerebral cortex [41]. The inflammatory activation index (TNFa to IL-10 ratio) shows higher and unresolved inflammation in the brains of AD mice before and after induction of periodontitis, than in non-AD mice; this is associated with a reduction in microglial markers in the vicinity of Aβ plaques, vs. those without induced periodontitis [26].

Consistent with the study by Kamer et al., which showed a higher expression of antibodies against periodontal bacteria (*A. actinomycetemcomitans*, *T. forsythia*, and *P. gingivalis*) and TNF-α in the serum of AD patients than controls [47]; Cestari et al. noted a significant association between periodontitis and increased serum levels of TNF-α and IL-6 in AD patients [48]. Over a six-month follow-up in patients with mild to moderate AD, anti-*P. gingivalis* antibody levels were associated with a fall in serum IL10 levels and an increase in serum TNFα [19]. In short-term post-mortem AD brain tissue (12h), Poole et al. (2013) detected the presence of Pg-LPS and suggests their role in brain inflammation associated with AD [49]. Similarly, Laugish et al. (2018) showed in patients with AD or other forms of dementia that high levels of antibodies against periodontal bacteria in cerebrospinal fluid and serum (*A. actinomycetemcomitans*, *T. denticola*, *T. forsythia*, *P. gingivalis*) can play a significant role in the local immune response [50]. In AD patients, a significant reduction in the concentration of sensitive biomarkers of immune activation is associated with different oral bacteria; thus, the presence of *T. denticola* and *T. forsythia* in saliva reduced the serum concentrations of neopterin and kynurenine, respectively [44].

### 3.7. Alzheimer’s Disease Risk Factor and Therapeutic Perspectives in Relation to the Oral Microbiome

The influence of genetics on the onset and development of AD, as well as on susceptibility to pathogens is well-understood [51,52]. In 2019, Liu et al. study found a significant decrease in Actinobacillus and Actinomyces levels and an abundance of Abiotrophia and Desulfomicrobium in patients carrying the APOE ε4 gene [34]. In *P. gingivalis*-infected mice, the expression of APP and BACE1 genes, involved in the development of AD, was significantly increased vs. the control group. Conversely, the expression of ADAM10, which plays an essential role in reducing the generation of amyloid-β (Aβ) peptides, was significantly reduced [37]. The Mendelian randomization study of Sun et al. (2020) found no evidence for a bidirectional genetic relationship between AD and periodontitis from analysis of data from Genome-Wide Association Studies in the European ancestry population [53]. According to Kamer et al., no significant difference in cytokine levels could be found as per the presence of the APOE ε4 gene in AD patients [47].

Several studies suggest that antimicrobial therapy is effective for AD [54], but only one study on modulation of the oral microbiome was identified. In the study by Dominy et al., oral administration of gingipain inhibitors to mice, essential for the survival of *P. gingivalis*, reduced the load of *P. gingivalis* and cytokines (TNFα), as well as the production of Aβ peptide 1–42 in the brain. An increased number of neurons in the hippocampus of *P. gingivalis*-infected mice, treated with these inhibitors, reflects a reduction in the neurotoxicity of this bacterium [39].

## 4. Discussion 

### 4.1. Oral Microbiome and Alzheimer’s Disease

The bidirectional relationship between the gut microbiome and AD is increasingly well-understood, leading to the definition of the gut–brain axis [55]. The detection of peptides in the gut and brain helped to lay the foundation for the concept of the gut–brain axis in the sixties and seventies [56]. Similar studies argue for a relationship between the oral microbiome and AD. Thus, the particular dysbiotic signature in Alzheimer’s patients involving periodontal bacteria (*A. actinomycetemcomitans*, *P. gingivalis*, *T. denticola*, *F. nucleatum)* suggests a modulation of oral bacterial flora by the central nervous system as observed for intestinal flora [57]. This finding must be moderated, due to additional factors affecting the oral microbiome of AD patients. Poor oral hygiene from reduced cognition and dexterity, age-related xerostomia, and medication can contribute to oral dysbiosis in these patients [58].

The presence of *P. gingivalis* in the brain, as well as an increase in Aβ peptide secretion, induced by associated compounds (gingipain, phosphoglycerol dihydroceramide), can be related to the antimicrobial activity of Aβ peptide, as demonstrated by Soscia et al. against several Gram-negative and Gram-positive bacteria (*Esherichia coli*, *Streptococcus pneumonia*, *Streptococcus salivarius*) as well as *Candida albicans* [59]. Long-term colonization by pathogenic bacteria and failure to clear the Aβ peptides, which was produced in response, can lead to excessive accumulation followed by brain tissue destruction through the formation of amyloid plaques and hyperphosphorylation of the tau protein [60]. These observations can be related to the survey results by Beydoun et al. [32], identifying a correlation between periodontal bacterial communities and the incidence or mortality of AD. In this context, while the evidence for the impact of the microbiome on cognitive performance seems well established [61], the results available in this review regarding the involvement of the oral microbiome on cognitive decline in AD patients seems contradictory.

### 4.2. Possible Mechanisms of the Relationship between the Oral Microbiome and Alzheimer’s Disease

Neuroinflammation is an important process in the neurodegeneration associated with AD. It exacerbates key features of the disease, including the deposition of Aβ peptides and hyperphosphorylation of tau. It places this process in a vicious cycle of inflammation and cellular destruction [62]. Thus, the presence in the brain of lipopolysaccharides from *T. denticola*, *T. forsythia*, and *P. gingivalis* that can be associated with increased expression of major proinflammatory cytokines (IL-1β, IL-6, TNFα, and IFNγ) and decreased anti-inflammatory cytokines (IL-10) suggests an involvement of the oral microbiota in the occurrence and progression of AD. Furthermore, these periodontal pathogens seem to modulate the immune response in AD patients. This modulation may be related to an alteration of microglial activity associated with an increased risk of AD [63]. However, localized transient inflammation is the normal immunological response of our immune system to cell/tissue damage, and is very important for tissue regeneration and repair [64]. Inflammation is a violent cell self-destructive way of clearing damaged cells. By destroying the infected tissues, the pathogens are cleared as well, and thus almost all microbial infections are self-limiting [65,66]. If at the same time the debris of the destroyed cells can be effectively cleaned away by our immune system [67], for example through autophagy, tissue regeneration will be followed, and infection-induced tissue damage will be transient and not become chronic, the patient can easily recover from such transient tissue damage.

While several mechanisms involving nervous, endocrine, and immune signals were advanced in the context of the gut–brain axis [57], no study identified mechanisms of the interaction between oral bacteria and AD. Nevertheless, Rivière et al. proposed the hypothesis that oral bacteria may infect the brain via branches of the trigeminal nerve [36]. The possibility of oral bacteria accessing the bloodstream [68] is a risk for sepsis [69,70] and may be associated more rarely with meningitis when these bacteria translocate to the brain [71,72]. It can be assumed that their interactions with the blood–brain barrier may also alter its integrity, allowing metabolic products produced by the microbiome, such as short-chain fatty acids, to cross the BBB and then affect brain function, such as the proposed in the gut–brain axis [73]. Although alterations in the BBB have been found both in AD patients and in animal models, their role in the disease process remains unclear [74].

The reciprocal modulation of the expression of some genes involved in AD and the modulation of oral microbiota has been identified (34,37). It provides the basis for the hypothesis of a plausible common genetic cause for oral dysbiosis and AD. However, according to Sun et al. [53], this hypothesis must be balanced due to the lack of evidence for a bidirectional genetic relationship between AD and periodontitis. The oral microbiome and associated dysbiosis appear to be only one environmental risk factor for the expression of genes associated with AD, as is the case for diseases such as diabetes or cardiovascular disease, etc. [75].

### 4.3. Inflammation, Overnutrition and Lipotoxicity

Although the virulence of the microorganism determines the damages made by the microorganisms to the host cells before they are cleared by host immunity, this microbial virulence alone does not account for the virulence of disease [66,76]. Most of the virulence of an infectious or chronic non-infectious disease is the result of the overactive inflammation response of our immune system [66,76], which turns local transient inflammation into systemic or chronic inflammation. The driving force for such an inflammatory transition is overnutrition. This is because, when there is overnutrition, the debris of the destroyed cells cannot be effectively cleaned, then this debris will become the nutrition source for bacteria proliferation. This debris can also be turned into lipid intermediates, and induce lipotoxicity. The human immune system also has a pivotal role in nutrition acquisition from the pathogen or the commensal microbiota like bacteria and the damaged tissues [77], and uses them for tissue regeneration if these nutrients just meet the need of tissue regeneration without surplus.

But if there is overnutrition, lipotoxicity as a result of overnutrition will prevent the tissue healing process from happening. Moreover, in the state of over-nutrition, the nutrition generated from the degradation of microbial-infected tissues will make this overnutrition situation worse, lipotoxicity and tissue damage will form a vicious positive feedback loop (as lipotoxicity is also a strong stimulus for cell dysfunction and cell death), which are escalated by the inflammation response, shifting inflammation from local to systemic, creating collateral damage to all other organs including the central nervous system, and may cause AD. So overnutrition could be a more possible linking factor between microbiota (oral or gastrointestinal) and AD.

### 4.4. Oral Microbiota Modulation and Alzheimer’s Disease Therapy

Treatments for AD are currently symptomatic, but research is targeting its pathological features, including Aβ peptides and tau protein [78]. In this context, the probable impact of the oral microbiome on the development of AD opens the door to a complementary therapeutic approach. Only the study by Dominy et al. (2019) is in line with this research by showing the interest in gingipain inhibitors in AD [39]. However, phototherapy, as well as prebiotic compounds and probiotic strains for a global/targeted modulation of the oral microbiome, represent areas of research for the prevention and treatment of AD. Indeed, the results of these potential new methods seem to demonstrate the effectiveness in the modification of the ecosystem of dental plaque by reducing the importance of the periodontal bacteria, which appear to be associated with AD (*P. gingivalis*, *F. nucleatum*, *C. rectus*, and others) [79,80]. Blue light is painless, fast, without drug toxicity or effect on taste, and is selective in effect. Its use could help to reduce oral dysbiosis linked to PD [81] and its impacts on AD patients. Probiotic strains affecting the periodontogenic flora through various galenic forms, such as milk drinks, yoghurts, and mouthwash could facilitate oral hygiene measures in AD patients [82].

Modulation of several pathological events in AD such as reduction in amyloid-β aggregation and inflammation, regulation of mitochondrial dynamics and increased availability of neuronal energy seem to be associated with different metabolic pathways (e.g., Wnt signalling, 5’-adenosine monophosphate-activated protein kinase) [83]. As Gram-negative bacteria-associated LPS and microbiome-generated amyloids potentially contribute to the regulation of these signalling pathways [84,85], therapeutic strategies to modulate oral dysbiosis as well as to modify bacterially produced amyloids or reduce their production also represent a future focus of research for the treatment of AD.

## 5. Limitations and Perspectives

While the relationship between the oral microbiome and mental disorders is a topic of growing interest, some limitations in the studies in this review must be noted. Thus, in the context of the evaluation of brain colonization by oral bacteria, some authors point to a risk of tissue contamination by the oral microbiota post-mortem [36]. Moreover, despite the microbiome changes over the life course [86], studies are predominantly based on case-control surveys and relatively small samples that do not allow assessment of the impact of exposure duration on the risk of developing AD. The promotion of longitudinal studies is essential to understand the role of the oral microbiome in the incidence and progression of AD.

The decline in cognitive function in AD patients is linked to behavioural changes that affect oral health. The progressive decline in patients’ ability and interest in both performing brushing and complementary oral care leads to the rapid development of oral pathology and dysbiosis [87]. In addition to difficulties in dental self-care, access to professional dental care procedures is complicated by a progressive cognitive decline in AD patients [88]. Faced with the challenge of an aging population and the increase in the weight of associated AD [89], it appears important to develop research towards solutions that promote the maintenance of the oral flora balance and to raise awareness among health professionals on the prevention of oral dysbiosis.

In addition, with genetic tools and molecular analyses available to identify individual bacteria and their metabolic products, it is possible to locate and track these bacteria in different host tissues to understand which cells they may affect and which metabolic signaling pathways they may activate and thus better understand how AD may be influenced by the oral microbiome. Due to the complexity of the etiopathology of AD and the interrelation between the factors responsible for the symptoms of the disease, a new therapeutic approach combining different targets is essential to develop new, more effective and personalized interventions and strategies for patients with this disease [90].

## 6. Conclusions

Several studies show a statistical link between AD and oral microbiota. Although the robustness of this link must be confirmed due to numerous confounding factors, evidence is accumulating towards the impact of the oral microbiome in pathophysiological and immuno-inflammatory mechanisms of this disease. The impact of the oral microbiome on the development of AD opens up complementary therapies, such as phototherapy, prebiotic compounds, and probiotic strains for a global or targeted modulation of the oral microbiome, with a favourable influence on the evolution of this pathology.

## Figures and Tables

**Figure 1 ijerph-18-11157-f001:**
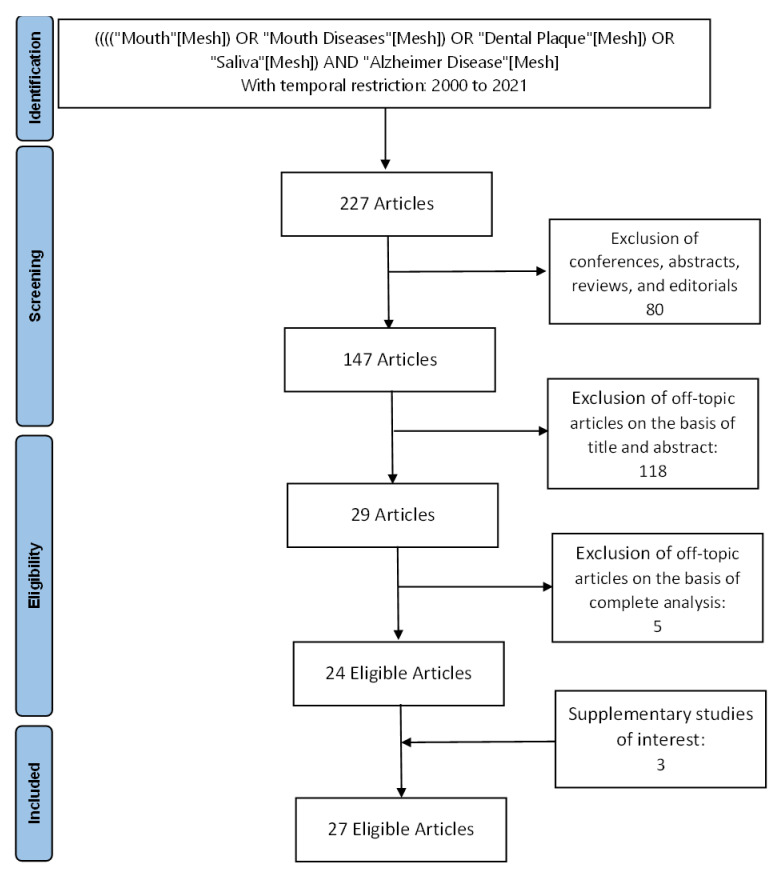
Flow chart of the study selection.

**Figure 2 ijerph-18-11157-f002:**
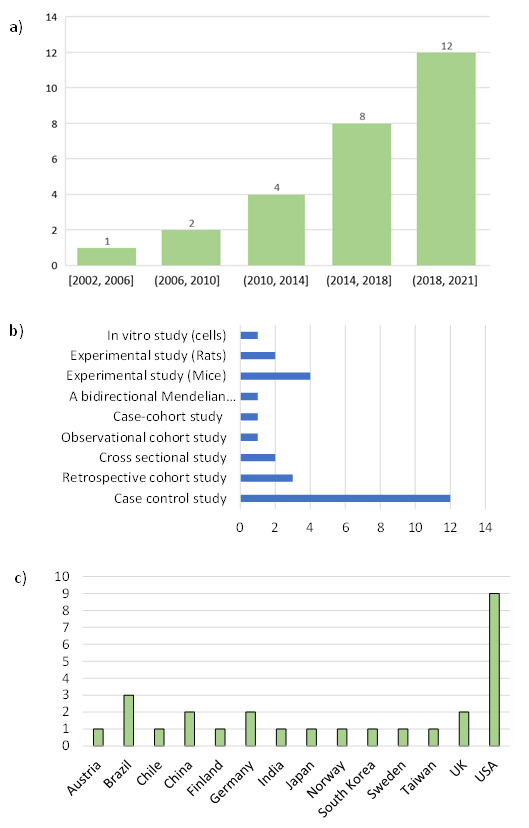
Study characteristics: years of publication (**a**), design, (**b**), research laboratory country (**c**).

**Table 1 ijerph-18-11157-t001:** Analysis of selected articles.

Reference(s)	Study Design	Objectives	Evaluated Factors	Mains Results and Limitations *
Riviere et al., 2002 [36]	Case control study N = 38/40	Assessment of the oral treponema presence in the human brain, the hippocampus and the trigeminal ganglia in AD patients and controls.	*T. denticola*, *Treponema amylovorum*, *Treponema maltophilum*, *T. medium*, *Treponema pectinovorum*, *Treponema socranskii*, *T. vincentii*	Detection of oral Treponema in brainstem, cortex and trigeminal ganglia in human subjects with Alzheimer’s disease. AD patients had more different Treponema species in the brain than controls. Treponema brain colonisation significantly more frequent in patients with Alzheimer’s disease than in controls. * Possibility of a post-mortem contamination of tissues or assays
Kamer et al., 2009 [47]	Case-control study N = 18/16	Evaluation of the difference in concentration of TNF α and antibodies to periodontal bacteria in serum of AD and normal controls. Investigation of the use of biomarkers (TNF α and antibodies against periodontal bacteria) for the diagnosis of AD)	*A. actinomycetemcomitans*, *T. forsythia*, *P. gingivalis* (IgG). TNF-α, IL-1β and IL-6 in plasma. APOE 4ε gene expression.	Higher expression of antibodies to periodontal bacteria and TNF α in serum of AD patients than in controls. These measures could contribute to the diagnosis of AD. No significant difference in cytokine levels between carriers of APOE 4ε and non-carriers * Small sample size, study design
Syrjälä et al., 2010 [29]	Case control study N = 76/278	Analysis of the association between diagnosed dementia and oral health among an elderly population aged 75 years or older.	Diagnosis of dementia as Alzheimer’s disease, vascular dementia, dementia due to other general medical (DSM-IV), dementia with Lewy bodies (McKeith Criteria). Number of teeth with dental caries and with deep periodontal pockets (≥4mm), edentulousness, oral hygiene index.	Significant probability to have carious teeth, teeth with deep periodontal pockets and poor oral and denture hygiene in patient with Alzheimer’s disease and persons with other types of dementia compared with non-demented persons. * Study design, sample size, lack of an assessment of inter-examiner and intra-examiner reliability of the oral examinations.
Poole et al., 2013 [49]	Case control study N = 10/10	Assessment of the presence of the major periodontal bacteria and/or their components in the brain tissue of people with and without dementia.	*P. gingivalis*, *T. denticola*, *T. forsythia*	Presence of *P. gingivalis*- LPS in AD brain with 12 h maximum postmortem delay. No evidence of *P. gingivalis* LPS in brain tissue of non-AD controls with a longer post-mortem time (up to 43 h). * Study design, small sample size.
De Souza Rolim et al., 2014 [28]	Case control study N = 29/30	Assessment of mandibular function, orofacial pain and oral status in patients with mild AD compared to age- and sex-matched healthy subjects.	Temporomandibular disorders McGill Pain Questionnaire, DMFT index, plaque and gingival bleeding indexes (PI, BI), PPD, cementoenamel junction (CEJ) and (CAL)	Significant higher prevalence of orofacial pain, articular abnormalities in temporomandibular joints and periodontal infections in AD patients than in healthy controls. * Study design, sample size
Martande et al., 2014 [31]	Case control study N = 58/60	Comparison of periodontal health status in people with and without AD from 50 to 80 years	Plaque index (PI), gingival index (GI), PPD, CAL), percentage bleeding on probing (% BOP). Degree of cognitive impairment by Mini-Mental State Examination (MMSE)	Significant differences in mean GI, PI, PD, CAL, and %BOP between different groups (Non AD, Mild, moderate and severe AD). Significant elevation of periodontal parameters in AD patients compared to non-AD patients. Deterioration of periodontal status with AD progression * Study design, sample size, no assessment of socio-economic status, education, changes in oral hygiene regimes as dementia progresses, dietary changes and access to dental care.
Noble et al., 2014 [38]	Case-cohort study N = 110/109	Study of the association between pre-morbid levels of serum IgG antibodies to selected periodontal microbiota and risk for incident AD.	*P. gingivalis*, *T. forsythia*, *A. actinomycetemcomitans*, *T. denticola*, *C. rectus*, *Eubacterium nodatum*, *Actinomyces naeslundii* (IgG)	The serum concentration of immunoglobulin G (IgG) against common periodontal bacteria represents a risk factor for developing AD. High concentrations of serum IgG associated to *Actinomyces naeslundii* was associated to an increased risk for developing incident AD. * Sample size, Possibility of reverse causality due to the lack of exclusion of patients with mild cognitive impairment (levels of antibodies to the periodontal microbiota may have been affected by cognitive impairment).
Kamer et al., 2015 [15]	Cross sectional study N = 38	Evaluation of the association between periodontal disease burden with the brain amyloid load in cognitively normal patients.	CAL, PPD, Bleeding on probing (BOP). Amyloid load in AD vulnerable brain areas (prefrontal cortex, middle frontal gyrus, lateral temporal lobe, inferior parietal lobule, and posterior cingulate cortex/precuneus)	Significant association between CAL and an increased amyloid load in vulnerable AD brain areas after adjusting for confounders (age, ApoE and smoking). No significant interaction between ApoE and periodontal measures. * Sample size, Limited generalisation of results due to the homogeneity of the study population, self-selected patient.
Cestari et al., 2016 [48]	Case control study N = 25/19/21	Investigation of the prevalence of oral infections and blood levels of IL-1β, TNF-α, and IL-6 in patients with AD, mild cognitive impairment (MCI), and onn demented controls.	Serum level of IL-1β, IL-6, and TNF-α.	Patients with AD had significantly higher IL-6 levels than controls. Significant association between IL-6 and TNF-α in patients with AD or MCI and periodontitis *Study design, sample size
Ide et al., 2016 [19]	Observational cohort study N = 60	Evaluation of the effects of periodontitis in patients with Alzheimer’s disease over a six-month follow-up.	*P. gingivalis* antibodies CRP, TNFα, IL 10 Cognitive scores (sMMSE and ADAS-COG)	No significant relationship between serum baseline *P. gingivalis* antibody levels and rates of cognitive decline. *P. gingivalis* antibody levels were associated with a fall in serum IL10 levels and an increase in serum TNFα levels over a six month follow up. * Small sample size and short term follow up
Chen et al., 2017 [33]	Retrospective cohort study N = 27 963	Assessment of the risk of developing AD in patients with chronic periodontitis.	Chronic periodontitis and AD diagnosis (ICM 9).	No significant higher risk of developing AD in Patients with 1 year of CP exposure than those without CP. Significant higher risk of developing AD in Patients with 10 years of CP exposure than those without CP. Higher prevalence of hyperlipidaemia, depression, traumatic brain injury and co-morbiditiess in patients with CP had a than those in the unexposed cohort. * Possible underestimation of the incidence of CP or AD, No data regarding the severity of AD and education level.
Holmer et al., 2018 [30]	Case control study N = 154/76	Assessment of the possible increase risk of mild cognitive impairment (MCI), subjective cognitive decline (SCD) and Alzheimer’s disease (AD) associated with periodontal disease. Investigation of the potential associations between common biofilm-induced dental diseases (dental caries and endodonticula), their sequelae (tooth loss) and cognitive impairment.	Periodontal status (oral hygiene, PPD, bleeding on pocket probing (BoP), suppuration, tooth mobility furcation involvment, marginal alveolar bone loss (MABL). Number of teeth, dental implants present, and dental caries.	Generalized marginal alveolar bone loss is associated with early cognitive impairment and AD. Poor oral health was more prevalent among cases than among healthy controls. * Study design, sample size, possible temporal bias with reverse causality (reduced cognitive function leads to poor oral health), no APOE genotyping
Ilievski et al., 2018 [37]	Experimental study (mice) N = 20	Assessment of the neuropathological effects (production of extracellular Aβ42 and neurofibrillary tangles) of repeated chronic exposure to a periodontal pathogen.	*P. gingivalis*/Gingipain TNFα, IL1β, and IL6 Alzheimer’s disease-related genes expression (APP, BACE 1, PSEN1, ADAM10)	*P. gingivalis* /gingipain were detected in the hippocampus of mice infected with Pg. Significantly higher expression of the proinflammatory cytokine IL1β, IL6 and TNFα in the hippocampus of experimental mice than controls. Significant increase in the expression of APP and BACE1 in the experimental group compared to the control group. Significant decrease in the expression of ADAM10 in the experimental group compared to the control group. Detection of Phospho-Tau protein in the experimental but not in the control mice * No direct or indirect mechanisms to explain these changes could be identified.
Laugisch et al., 2018 [50]	Case control study N = 20/20	Verification of the presence of periodontal pathogens and the intrathecal generation of pathogen-specific antibodies in patients with AD and with other forms of dementia (DEM-noAD).	Antibody levels against *A. actinomycetemcomitans*, *T. denticola*, *T. forsythia*, *P. gingivalis* in CSF and serum Total tau protein (T-tau) and amyloid-β (Aβ1-42) in CSF. Monocyte chemoattractant protein-1 (MCP-1/CCL2)	Periodontal pathogens may enter the brain and stimulate a local immune response. Significant association between T-tau levels in the AD group and serum levels of anti-*P. gingivalis* antibodies and MCP-1/CCL-2. * Sample size
Maurer et al., 2018 [35]	Case control study N = 20/20	Evaluation of the possible link between Alzheimer’s dementia and bacterial infestation of the oral cavity.	*A. actinomycetemcomitans*, *P. gingivalis*, *F. nucleatum*	AD-patients showed higher bacterial load in dental plaque compared to controls. * Study design, sample size
Choi et al., 2019 [21]	Retrospective cohort study N = 262 349	Determination of the association between chronic periodontitis (CP), AD and vascular dementia (VD) from the Korean National Health Insurance Service (NHIS) database, using a several covariates (smoking, alcohol consumption and physical activity).	CP diagnosis (CD-10 code K05.3) associated with at least one of the CP-related treatments. Prescription of dementia-related drugs as part of a diagnosis of AD (ICD-10 codes F00, G30) or VD (ICD-10 code F01) Age, sex, household income, smoking status, alcohol consumption, physical activity, body mass index, systolic blood pressure, fasting serum glucose, total cholesterol and Charlson Comorbidity Index	Chronic periodontitis patients had elevated risk for overall dementia (aHR = 1.06; 95% CI = 1.01–1.11, *p* = 0.042) and AD (aHR = 1.05; 95% CI = 1.00–1.11, *p* = 0.042) than non-chronic periodontitis participants. * Study design, Information on CP-related clinical index limited, definition of dementia based on drug reimbursement, no evaluation of level of education or apolipoprotein E (APOE) e4 genotype.
Dominy et al., 2019 [39]	Experimental study (mice) N = 140	Assessment of the prevalence of *P. gingivalis* (Pg) in the brains of people with Alzheimer’s disease and possible Pg-dependent mechanisms of action in neurodegeneration and Alzheimer’s pathology.	*P. gingivalis* and Gingipaïn load in brain tissue	*P. gingivalis* and gingipains load in the brain play a significative role in the pathogenesis of AD. Reduction in *P. gingivalis* brain bacterial load and neuroinflammation as well as a blockage of Aβ 1–42 peptide production were obtained by gingipain inhibition. * Study design, sample size
Hayashi et al., 2019 [43]	Experimental study (Mice) N = 80	Evaluation of the effects of brain exposure to *Porphyromonas gingivalis*-derived lipopolysaccharide (Pg-LPS) on cognitive impairment and organ dysfunction in an AD mouse model.	*P. gingivalis* LPS Morris water maze tests	No cognitive impairment could be associated with acute or continuous brain exposure to Pg-LPS. Continuous brain exposure to Pg-LPS triggered sarcopenia and heart damage in AD model mice. * Study design
Liu et al., 2019 [34]	Case control study N = 39/39	Identification of differences in oral bacterial community composition between AD patients and healthy patients. Evaluation of the association between oral bacteria and AD severity Evaluation of differences in oral bacterial flora according to APOEε4 expression.	Alpha and beta diversity of salivary microbiota APOEε4 expression	Significant decrease of richness and diversity of salivary microbiota in patients with Alzheimer’s disease than in healthy controls. No bacteria associated with the severity of AD. Significant decrease in Actinobacillus and Actinomyces levels in patients with APOEε4 Abundance of Abiotrophia and Desulfomicrobium levels in patients with APOEε4(+) * Small sample size, no collection of dental plaque bacteria
Beydoun et al., 2020 [32]	Retrospective cohort study N = 6650	Evaluation of the association of immune response (IgG) to periodontal pathogens with the incidence of dementia and AD mortality in middle-aged (>45 years) and elderly (>65 years) US adults.	Periodontal pathogens Immunoglobulin G (IgG): *A. actinomycetemcomitans*, *P. gingivalis*, *T. forsythia*, *T. denticola*, *C. rectus*, *Eubacterium nodatum*, *P. intermedia*, *Prevotella nigrescens*, *Prevotella melaninogenica*, *F. nucleatum*, *Parvimonas micra*, *Selenomonas noxia*, *Eikenella corrodens*, *Capnocylophaga ochracea*, *Streptococcus intermedius*, *Streptococcus oralis*, *Streptococcus mutans*, *Vellonella Parvula*, *Actinomyces naeslundii*. AD Mortality and Incidence Status. Clinical Attachment loss (CAL) and probing pocket depth (PPD)	AD incidence was linked to a composite of *C. rectus* and *P. gingivalis.* AD mortality risk was increased with composite loading highly on IgG for *P. gingivalis*, *P. intermedia*, *Prevotella nigrescens*, *F. nucleatum*, *C. rectus*, *Streptococcus intermedius*, *Capnocylophaga Ochracea*, and *P. melaninogenica*. Only a marginal association between incident AD risk and PPD was detected among men and older individuals * Study design
De Oliveira Araújo et al., 2021 [27]	Case control study N = 50/52	Test the association between periodontitis and AD. Assessment of the possible negative impact of periodontal status on perceived oral health-related quality of life (OHRQoL)	PPD ≥ 5 mm, and CAL ≥ 5 mm Geriatric Oral Health Assessment Index (GOHAI) questionnaire. Socio-demographic data.	AD patients had fewer teeth, greater and a superior percentage of sites with plaque, calculus, and bleeding on probing than healthy controls. Significant association between periodontitis and AD after adjusting for age, gender, income and education. Periodontitis is associated with AD, but not with patients’ OHRQoL * Study design, sample size
Díaz-Zúñiga et al., 2020 [42]	Experimental study (Rats) N = 30	Evaluation of the effects of short exposure to encapsulated strains of *P. gingivalis* on AD brain markers, neuroinflammation and cognitive decline in young rats.	*P. gingivalis* K1, K2, or K4 serotypes and the K1-isogenic non-encapsulated mutant (GPA). Oasis maze task. Cytokines (IL-1b, IL-4, IL-6, IL-10, TNF-α, IFN- γ). Aβ1-42 peptide and tau phosphorylation levels in hippocampus.	Significant increase of pro-inflammatory cytokines (IL-1b, IL-6, TNF-a, IFN-γ) in the hippocampus of rats infected with *P. gingivalis* encapsulated serotypes K1 and K2. None of these effects were observed in rats infected with the non-encapsulated bacterial strains. K1 or K2 *P. gingivalis*-infected rats displayed memory deficits, increased Aβ 1–42 levels, and Tau hyperphosphorylation in the hippocampus * Sample size, study design
Kantarci et al., 2020 [26]	Experimental study (Mice) N = 30	Evaluation of the impact of experimentally induced periodontal disease in a mouse model of AD on the inflammatory process in the brain and microglia function.	Alveolar bone loss Insoluble Aβ40 and Aβ442 peptide. Microglial markers in brain (Iba1). Cytokine and Chemokine in CSF (GM-CSF, IFN-γ, IL-1β, IL-6, IL-10, TNF-α, MCP 1).	Periodontal disease increases bone loss in AD-modelled (5xFAD) and control wild-type (WT) mice. Alveolar bone loss is higher in 5xFAD than in WT at baseline. No significant difference in alveolar bone loss after the induction of experimental PD between 5xFAD and WT mice The mean level of insoluble Aβ42, but not Aβ40, was significantly higher in 5xFAD mice with induced PD than in 5xFAD mice without induced PD. Decline in microglial markers (Iba1 )in the proximity of Aβ plaques in 5xFAD mice with periodontal disease compared to those without periodontal disease. Induced periodontal disease reduced IL-10 in 5xFAD mice. Higher unresolved inflammation in the brain of 5xFAD mice before and after induction of periodontal disease compared to WT controls based on the ratio of TNF-α to IL-10. * Study design, Sample size, no characterization of the bacterial flora
Leblhuber et al., 2020 [44]	Cross sectional study N = 20	Evaluation of the effects of chronic low-grade immune activation by salivary periodontopathogen bacteria in AD patients	*A. actinomycetemcomitans*, *T. denticola*, *T. forsythia*, *P. gingivalis*, *P. intermedia* in salivary. MMSE and CDT. Serum concentrations of neopterin and of tryptophan.	Significant association between the salivary presence of *P. gingivalis* and lower MMSE and lower tendency to CDT. Significant lower neopterin concentrations associated with the presence of *T. denticola* in AD patient saliva. Significant lower kynurenine concentrations associated with the presence of *T. forsythia* in AD patient saliva. * Sample size, study design, no evaluation of the ApoE status
Sun et al., 2020 [53]	A bidirectional Mendelian randomization study N = 4924 /7301	Examination of the potential causal relationship between AD and chronic periodontitis bidirectionally in the population of European ancestry.	Single-nucleotide polymorphisms associated with periodontitis and AD.	No evidence for a bidirectional genetic relationship between AD and periodontitis from analysis of Genome-Wide Association Studies data. * Mendelian randomisation does not assess the impact of the duration of periodontal disease on the risk of developing AD, The genetic tool used to define PD may not be suitable for detecting the causal link between PD and AD.
Yamada et al., 2020 [40]	In vitro study (cells)	Evaluation of the influence of phosphoglycérol dihydrocéramide (PGDHC) produced by *P. gingivalis* on hallmark findings in AD.	Concentrations of PGDHC, Aβ peptide, Protein Tau and senescence-associated secretory phenotype (β-galactosidase, cathepsin B, cysteine, TNF-α and IL-6 pro-inflammatory cytokines)	Exposure to PGDHC, but not to Pg-LPS, significantly enhances secretion of Aβ peptide in a dose-dependent manner from CHO-7WD101 cells in vitro ^1^. PGDHC also significantly Induces the Site-Specific Phosphorylation of protein Tau in a dose-dependent manner in SH-SY5Y2 cells compared to the control cells. PGDHC or Pg-LPS elevated expression of β-galactosidase, cathepsin B, cysteine, TNF-α and IL-6 pro-inflammatory cytokines in SH-SY5Y cells. * In vitro study
Zhang et al., 2020 [41]	Experimental study (Rats) N = 24	Examination of the association between oral health and cognition in humans and rats.	Morris Water Maze test Concentrations of Aβ1–40 peptide, TNF-α, IL-1, IL-6 and CRP in the hippocampus and the cerebral cortex	No significant differences in the Morris Water Maze test between AD rats and AD rats with induced periodontitis. Significant elevation of Aβ1-40 concentration in the cerebral cortex in AD rats with periodontitis than in AD rats. TNF-α levels in the hippocampus of the AD with periodontitis group were significantly higher than those of the AD and the control group IL-1 levels in the cerebral cortex of the AD with periodontitis group were significantly higher than those of the AD group and the control group. IL-6 levels in the hippocampus and the cerebral cortex of the AD with periodontitis groups were significantly higher than those of the AD and the control group. * Study design, sample size, no detection of periodontal pathogens in the brain.

* Limitations of the sutdy. PD: Periodontal disease; AD: Alzheimer disease, MMSE: mini mental state examination; CDT: clock drawing test; ADAS-cog: Alzheimer’s Disease Assessment Scale CSF: Cerebro-spinal fluid, DMFT: decayed, missing, and filled teeth; IL: Interleukin; TNFα: Tumor necrosis factor α; CRP C-reactive protein; CAL: Clinical Attachment loss; PPD: probing pocket depth; *A. actinomycetemcomitans*: *Aggregatibacter actinomycetemcomitans*, *P. gingivalis*: *Porphyromonas gingivalis*, *T. forsythia*: *Tannerella forsythia*, *T. denticola*: *Treponema denticola*, *C. rectus*: *Campilobacter rectus*, *F. nucleatum*: *Fusobacterium nucleatum*, *P. intermedia*: *Prevotella intermedia*. ^1^ CHO-7WD10: Chinese hamster ovary-7WD10 (CHO-7WD10) cells stably expressing human wild-type amyloid precursor protein 751(APP751WT); ^2^ SH-SY5Y: human neuroblastoma cells.

## Data Availability

No data were generated.

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
