# Peer review of "Evidence and Therapeutic Perspectives in the Relationship between the Oral Microbiome and Alzheimer’s Disease: A Systematic Review"

_ijerph, 2021, doi:10.3390/ijerph182111157_

Round 1
Reviewer 1 Report
Comments to “Evidence and Therapeutic Perspectives in the Relationship between the Oral Microbiome and Alzheimer's Disease: a Systematic Review”
In this manuscript, the authors carried out a systematic review relating the Oral Microbiome to Alzheimer's Disease (AD). And suggested some complementary therapies like phototherapy and/or the use of prebiotic compounds and probiotic strains to modulate the oral microbiome in order to create a favourable influence on the evolution of AD. This systematic review provides a lot of insightful, comprehensive and pioneering thinking on the pathology of AD, may shed some light on the elusive causes and the treatment of AD. This paper would be more entrancing if the authors can take the following minor revision points on human immunity, inflammation and nutrition generation, overnutrition/lipotoxicity into consideration:
- When talking about microbiota, we also need to address the human immunity, especially inflammation and tissue regeneration. As pointed out by the authors repeatedly that chronic or systemic inflammation are linked to a lot of diseases including AD. Yet, localized transient inflammation is the normal immunological response of our immune system to cell/tissue damage, and is very important for tissue regeneration and repair [1]. Inflammation is a violent cell self-destructive way of clearing damaged cells. By destroying the infected tissues, the pathogens are cleared as well, and thus almost all microbial infections are self-limiting [2,3]. If at the same time the debris of the destroyed cells can be effectively cleaned away by our immune system [4], for example through autophagy, tissue regeneration will be followed, and infection-induced tissue damage will be transient and not become chronic, the patient can easily recover from such transient tissue damage.
- Although the virulence of the microorganism determines the damages made by the microorganisms to the host cells before they are cleared by host immunity, this microbial virulence alone doesn’t account for the virulence of a disease [3,5]. The most part of the virulence of an infectious or chronic non-infectious disease is the result of the overactive inflammation response of our immune system [3,5], which turns local transient inflammation into systemic or chronic inflammation. And the driving force for such inflammatory transition is overnutrition.
- As pointed out by the authors, metabolic disorders like hypertension, stroke, hyper-cholesterolemia, diabetes, overweight status and obesity are well-established risk factors for AD. This is because, when there is overnutrition, the debris of the destroyed cells cannot be effectively cleaned, then these debris will become the nutrition source for bacteria proliferation. These debris can also be turned into lipid intermediates, and induce lipotoxicity. The Human immune system also has pivotal role in nutrition acquisition from the pathogen or the commensal microbiota like bacteria and the damaged tissues [6], and uses them for tissue regeneration if these nutrients just meet the need of tissue regeneration without surplus.
- But if there is overnutrition, lipotoxicity as a result of overnutrition will prevent the tissue healing process from happening. Moreover, in the state of over-nutrition, the nutrition generated from degradation of microbial-infected tissues will make this overnutrition situation worse, lipotoxicity and tissue damage will form a vicious positive feedback loop (as lipotoxicity is also a strong stimulus for cell dysfunction and cell death), which are escalated by the inflammation response, shifting inflammation from local to systemic, creating collateral damage to all other organs including the central nervous system, and may cause AD. So overnutrition could be a more possible linking factor between microbiota (oral or gastrointestinal) and AD.
On page 4, in “Study characterization”, the statement “with 44% of publications from 2009 to 2021 (Figure 2a)” is incorrect. A close check of the 27 articles included in this systematic review shows that only one paper was published in 2002, the other 26 papers were published during 2009 to 2021. In this regard, Figure 2(a) on page 5 also should be re-drawn to remove the overlapping years in the x-axis and drop out the year 2022.
On page 12, line 17 from bottom to line 16 from bottom, the authors cited Rivière et al. (Reference 36) proposed the hypothesis that “oral bacteria may infect the brain” and cited Logsdon et al. (reference 64) saying that “oral bacteria accessing the bloodstream”. Such hypotheses are very unlikely for patients with AD, because bacteria translocation to bloodstream is sepsis [7,8], and bacteria translocation to brain is meningitis [9], both are acute bacterial infections which are very different from the symptoms of AD which is chronic degradation of the nervous tissues.
The following related references may be included in this manuscript to provide a more complete picture on human immunity, microbiota, infection, inflammation, overnutrition and disease:
Reference:
- Cooke JP (2019) Inflammation and its role in regeneration and repair - a caution for novel anti-inflammatory therapies. Circ Res, 124:1166–1168. DOI: 1161/CIRCRESAHA.118.314669.
- Levin BR, Baquero F, Ankomah P, McCall IC (2017) Phagocytes, Antibiotics, and Self-Limiting Bacterial Infections. Trends in Microbiology, 25(11):878-892. DOI: 10.1016/j.tim.2017.07.005
- Levin BR, Antia R (2001) Why we don't get sick: The within-host population dynamics of bacterial infections. Science, 292:1112-1115. DOI: 10.1126/science.1058879.
- Cunliffe J (2007) Intentional pathogen killing–or denial of substrate? Scand J Immunol 66:604–9. DOI: 10.1111/j.1365-3083.2007.02017.x.
- Humphries DL, Scott ME, Vermund SH (2020) Pathways linking nutritional status and infectious disease. In: Humphries DL, Scott ME, Vermund SH, editors. Nutrition and infectious disease: shifting the clinical paradigm: Humana Press; pp4-5. DOI: 10.1007/978-3-030-56913-6_1.
- Yu BX, Yu LG & Klionsky DJ. Nutrition Acquisition by Human Immunity, Transient Overnutrition and the Cytokine Storm in Severe Cases of COVID-19. Medical Hypotheses, 155, 110668 (2021). DOI: 10.1016/j.mehy.2021.110668.
- Berg RD (1999) Bacterial Translocation from the Gastrointestinal Tract. In: Paul PS, Francis DH (eds) Mechanisms in the Pathogenesis of Enteric Diseases 2. Advances in Experimental Medicine and Biology, vol 473. Springer, Boston, MA. DOI: 10.1007/978-1-4615-4143-1_2.
- Rhodes A, Evans LE, Alhazzani W et al. (2017) Surviving Sepsis Campaign: International Guidelines for Management of Sepsis and Septic Shock: 2016. Intensive Care Med 43, 304–377. DOI: 10.1007/s00134-017-4683-6.
- Kim K.S. (2006) Microbial translocation of the blood–brain barrier. Int J Parasitol 36: 607–614. DOI10.1016/j.ijpara.2006.01.013.
Author Response
Responses to reviewer
Reviewer 1:
1/ In this manuscript, the authors carried out a systematic review relating the Oral Microbiome to Alzheimer's Disease (AD). And suggested some complementary therapies like phototherapy and/or the use of prebiotic compounds and probiotic strains to modulate the oral microbiome in order to create a favourable influence on the evolution of AD. This systematic review provides a lot of insightful, comprehensive and pioneering thinking on the pathology of AD, may shed some light on the elusive causes and the treatment of AD.
Response
Thank you.
2/ This paper would be more entrancing if the authors can take the following minor revision points on human immunity, inflammation and nutrition generation, overnutrition/lipotoxicity into consideration:
When talking about microbiota, we also need to address the human immunity, especially inflammation and tissue regeneration. As pointed out by the authors repeatedly that chronic or systemic inflammation are linked to a lot of diseases including AD. Yet, localized transient inflammation is the normal immunological response of our immune system to cell/tissue damage, and is very important for tissue regeneration and repair [1]. Inflammation is a violent cell self-destructive way of clearing damaged cells. By destroying the infected tissues, the pathogens are cleared as well, and thus almost all microbial infections are self-limiting [2,3]. If at the same time the debris of the destroyed cells can be effectively cleaned away by our immune system [4], for example through autophagy, tissue regeneration will be followed, and infection-induced tissue damage will be transient and not become chronic, the patient can easily recover from such transient tissue damage.
Although the virulence of the microorganism determines the damages made by the microorganisms to the host cells before they are cleared by host immunity, this microbial virulence alone doesn’t account for the virulence of a disease [3,5]. The most part of the virulence of an infectious or chronic non-infectious disease is the result of the overactive inflammation response of our immune system [3,5], which turns local transient inflammation into systemic or chronic inflammation. And the driving force for such inflammatory transition is overnutrition.
As pointed out by the authors, metabolic disorders like hypertension, stroke, hyper-cholesterolemia, diabetes, overweight status and obesity are well-established risk factors for AD. This is because, when there is overnutrition, the debris of the destroyed cells cannot be effectively cleaned, then these debris will become the nutrition source for bacteria proliferation. These debris can also be turned into lipid intermediates, and induce lipotoxicity. The Human immune system also has pivotal role in nutrition acquisition from the pathogen or the commensal microbiota like bacteria and the damaged tissues [6], and uses them for tissue regeneration if these nutrients just meet the need of tissue regeneration without surplus.
But if there is overnutrition, lipotoxicity as a result of overnutrition will prevent the tissue healing process from happening. Moreover, in the state of over-nutrition, the nutrition generated from degradation of microbial-infected tissues will make this overnutrition situation worse, lipotoxicity and tissue damage will form a vicious positive feedback loop (as lipotoxicity is also a strong stimulus for cell dysfunction and cell death), which are escalated by the inflammation response, shifting inflammation from local to systemic, creating collateral damage to all other organs including the central nervous system, and may cause AD. So overnutrition could be a more possible linking factor between microbiota (oral or gastrointestinal) and AD.
Response
We would like to thank the reviewer for his valuable suggestions, which have made it possible to significantly improve our work. We have added a new paragraph on page 14, lines 304 to 313, and a new chapter entitled "4.3. Inflammation, overnutrition and lipotoxicity" on page 15.
3/ On page 4, in “Study characterization”, the statement “with 44% of publications from 2009 to 2021 (Figure 2a)” is incorrect. A close check of the 27 articles included in this systematic review shows that only one paper was published in 2002, the other 26 papers were published during 2009 to 2021. In this regard, Figure 2(a) on page 5 also should be re-drawn to remove the overlapping years in the x-axis and drop out the year 2022.
Response
Thank you for pointing out this error. Indeed, 44% of the studies were recently published between 2019 and 2021 and not between 2009 and 2021. This error has been corrected in the manuscript, please see line 124.
We have retained the proposed chronological breakdown to highlight the increase in interest in the topic over time. As suggested, we have removed the year 2022 in Figure 2a to be more accurate.
4/ On page 12, line 17 from bottom to line 16 from bottom, the authors cited Rivière et al. (Reference 36) proposed the hypothesis that “oral bacteria may infect the brain” and cited Logsdon et al. (reference 64) saying that “oral bacteria accessing the bloodstream”. Such hypotheses are very unlikely for patients with AD, because bacteria translocation to bloodstream is sepsis [7,8], and bacteria translocation to brain is meningitis [9], both are acute bacterial infections which are very different from the symptoms of AD which is chronic degradation of the nervous tissues.
Response
We fully agree that the passage of microbes into the bloodstream or brain respectively represents sepsis or meningitis. As suggested, we underlying this point in the text.
We have clarified that as a result of alteration of the blood-brain barrier (BBB) by the microbiome, metabolic products produced by the microbiome can cross the BBB and impair brain function as suggested by Longsdon. However, we have moderated our comments by stating that although alterations in the BBB are found in AD patients and animal models, their role in the disease process remains unclear. Please see line 318 to 326.
Proposed references and the following reference have been added and listed in the references section:
- Berg, R.D. Bacterial Translocation from the Gastrointestinal Tract. Springer: Boston, United States of America, 1999; 11-50
- Rhodes A, Evans LE, Alhazzani W, Levy MM, Antonelli M, Ferrer R, Kumar A, Sevransky JE, Sprung CL, Nunnally ME, et al. Surviving Sepsis Campaign: International Guidelines for Management of Sepsis and Septic Shock: 2016. Intensive Care Med. 2017, 43,304-377.
71 Maurer P, Hoffman E, Mast H. Bacterial meningitis after tooth extraction. Br Dent J. 2009 Jan 24;206(2):69-71.
- Kim, K.S. Microbial translocation of the blood-brain barrier. Int J Parasitol. 2006, 36, 607-614.
- Montagne, A.; Zhao, Z.; Zlokovic, B.V. Alzheimer's disease: A matter of blood-brain barrier dysfunction? J Exp Med. 2017,214,3151-3169.
5/ The following related references may be included in this manuscript to provide a more complete picture on human immunity, microbiota, infection, inflammation, overnutrition and disease:
Reference :
- Cooke JP (2019) Inflammation and its role in regeneration and repair - a caution for novel anti-inflammatory therapies. Circ Res, 124:1166–1168. DOI: 1161/CIRCRESAHA.118.314669.
- Levin BR, Baquero F, Ankomah P, McCall IC (2017) Phagocytes, Antibiotics, and Self-Limiting Bacterial Infections. Trends in Microbiology, 25(11):878-892. DOI: 10.1016/j.tim.2017.07.005
- Levin BR, Antia R (2001) Why we don't get sick: The within-host population dynamics of bacterial infections. Science, 292:1112-1115. DOI: 10.1126/science.1058879.
- Cunliffe J (2007) Intentional pathogen killing–or denial of substrate? Scand J Immunol 66:604–9. DOI: 10.1111/j.1365-3083.2007.02017.x.
- Humphries DL, Scott ME, Vermund SH (2020) Pathways linking nutritional status and infectious disease. In: Humphries DL, Scott ME, Vermund SH, editors. Nutrition and infectious disease: shifting the clinical paradigm: Humana Press; pp4-5. DOI: 10.1007/978-3-030-56913-6_1.
- Yu BX, Yu LG & Klionsky DJ. Nutrition Acquisition by Human Immunity, Transient Overnutrition and the Cytokine Storm in Severe Cases of COVID-19. Medical Hypotheses, 155, 110668 (2021). DOI: 10.1016/j.mehy.2021.110668.
- Berg RD (1999) Bacterial Translocation from the Gastrointestinal Tract. In: Paul PS, Francis DH (eds) Mechanisms in the Pathogenesis of Enteric Diseases 2. Advances in Experimental Medicine and Biology, vol 473. Springer, Boston, MA. DOI: 10.1007/978-1-4615-4143-1_2.
- Rhodes A, Evans LE, Alhazzani W et al. (2017) Surviving Sepsis Campaign: International Guidelines for Management of Sepsis and Septic Shock: 2016. Intensive Care Med 43, 304–377. DOI: 10.1007/s00134-017-4683-6.
- Kim K.S. (2006) Microbial translocation of the blood–brain barrier. Int J Parasitol 36: 607–614. DOI10.1016/j.ijpara.2006.01.013.
Response
All suggested references have been added and listed in the references section. Thanks again for these valuable suggestions
Reviewer 2 Report
In this review article the author touched the important topic on correlation with Alzheimer’s disease (AD) and oral microbiota. They discussed on epidemiological, gene expression changes and therapeutic relevance. I have few comments and suggestion to improve the review article.
- In the abstract section, remove the blank link after “We included ex”.
- In the result section 3.1 add the full stop after the sentence “Of the 227 studies initially identified, 24 were selected.
- The quality of the figure is poor. The details are not readable. Please include high quality figures.
- In Figure 1 flow chart, as the author included three additional supplementary articles, it can be better represented with inward arrow, rather than outward arrow, which is confusing.
- In Figure 2, the figure numbering 2a, 2b and 2c should be placed in the top-left position of the figure.
In the discussion section, the author should discuss about the molecular advances and signaling pathways altered in AD.
Author Response
Responses to reviewer
Reviewer 2:
In this review article the author touched the important topic on correlation with Alzheimer’s disease (AD) and oral microbiota. They discussed on epidemiological, gene expression changes and therapeutic relevance. I have few comments and suggestion to improve the review article.
1/ In the abstract section, remove the blank link after “We included ex”.
Response
As suggested, we removed the blank link in the abstract section.
2/ In the result section 3.1 add the full stop after the sentence “Of the 227 studies initially identified, 24 were selected.
Response
As suggested, we added the full stop after the sentence “Of the 227 studies initially identified, 24 were selected”. (Line 115)
3/ The quality of the figure is poor. The details are not readable. Please include high quality figures.
Response
For better readability the writing has been done in black instead of grey. The pdf files of these figures have been submitted separately from the manuscript in order to benefit from a better definition than that available in the text.
4/ In Figure 1 flow chart, as the author included three additional supplementary articles, it can be better represented with inward arrow, rather than outward arrow, which is confusing.
Response
Thanks for the suggestion, Figure 1 has been modified for clarity.
5/ In Figure 2, the figure numbering 2a, 2b and 2c should be placed in the top-left position of the figure.
Response
The figure numbering 2a, 2b and 2c was placed as requested in Figure 2.
6/ In the discussion section, the author should discuss about the molecular advances and signaling pathways altered in AD.
Response
As suggested, these points were presented in the discussion section page 15 line 374 to 381 and page 16 line 401 to 408.
The following reference has been added and listed in the references section:
- Godoy, J.A.; Rios, J.A.; Zolezzi, J.M.; Braidy, N.; Inestrosa, N.C. Signaling pathway cross talk in Alzheimer's disease. Cell Commun Signal. 2014, 12, 23.
- Jiang, C.; Li, G.; Huang, P.; Liu, Z.; Zhao, B. The Gut Microbiota and Alzheimer's Disease. J Alzheimers Dis. 2017, 58, 1-15.
- Friedland, R.P.; Chapman, M.R. The role of microbial amyloid in neurodegeneration. PLoS Pathog. 2017, 13, e1006654.
88- Martínez-Cué, C.; Rueda N. Signalling Pathways Implicated in Alzheimer's Disease Neurodegeneration in Individuals with and without Down Syndrome. Int J Mol Sci. 2020, 21, 6906.